# Wikipedia, friend or foe regarding information on diabetic retinopathy? A content analysis in the world's leading 19 languages

Kouatzin Aguilar-Morales[1], Gustavo Aguirre-Suarez[1], Brigham Bowles[2,3], Angel Lee[2,4]*, Van Charles Lansingh[1,5]*

1 Instituto Mexicano de Oftalmología IAP, Colinas de Cimatario, Centro Sur, Santiago de Querétaro, Mexico, 2 Instituto Nacional de Neurología y Neurocirugía, La Fama, Ciudad de México, Mexico, 3 Universidad Westhill, Santa Fe Cuajimalpa, Ciudad de México, Mexico, 4 Comisión Coordinadora de Institutos Nacionales de Salud, Arenal Tepepan, Ciudad de México, Mexico, 5 Help Me See Inc, New York, New York, United States of America

* vancharles@helpmesee.org (VCL); dr_angel_lee@yahoo.de (AL)

**Data Availability Statement:** All relevant data are within the manuscript and its Supporting information files.

## Abstract

### Objective

To compare the completeness and quality of information about diabetic retinopathy on Wikipedia in the world's leading spoken languages in 2020.

### Design and methods

An observational, descriptive, cross-sectional study. The information on diabetic retinopathy obtained from the free encyclopedia Wikipedia® was assessed in languages with one hundred million or more total speakers. The term "diabetic retinopathy" was accessed in the corresponding Wikipedia entry in English, while the "more languages" function gives access to other languages. The information on the sites was collected by three ophthalmologist observers. A database was created with the most important subtopics for the education of patients with diabetic retinopathy in any of its classifications, based on a 25-question survey. The results were stratified on a scale from 0 to 4. A confirming correlation was found in the statistical analysis among the observers.

### Results

No language achieved the label "excellent"; 2 languages were rated as "fair "; 4 languages qualified as "substandard"; and 7 languages were scored as "poor." No information could be found in five languages.

### Conclusions

As would be expected, the quality of content is variable across different languages. However, if anyone can edit Wikipedia, health professionals can do so as well to improve the quality and quantity of information for patients.

**Funding:** The authors received no specific funding for this work.

**Competing interests:** The authors have declared that no competing interests exist.

## Introduction

The increase in the incidence of diabetes mellitus (DM) and the costs related to the treatment of its complications are a matter of global public health concern. It is projected that by 2030 there will be 578.4 million, and by 2045, 700.2 million adults aged between 20 and 79 years with this disease [1,2]. Diabetic retinopathy (DR) is one of the leading causes of blindness and poor vision worldwide [3]. The greatest increase is projected to take place in low- to middle-income regions [1,4]. which also exhibit higher complication rates [5]. It has been shown that personalized education and risk assessment during ophthalmology visits did not result in a reduction in HbA1c levels compared to usual care for one year [6]. As in many cases where information provided by doctors requires further explanation, one of the main sources of clarification for patients is the Internet. Some 80% of users report some searches related to health issues, and 53% of this group asserted that the information thus obtained had an impact on the care of themselves or a relative [6]. Patients indicate that they would search the Internet with more confidence if their doctor would recommend where to access the needed information [7]. The preferred site for patients to search for information on health issues is Wikipedia, which in 2018 alone had 2,194,804,393 visits [8]; thus it could be referred to as "the main source of medical information for patients and health professionals." In addition, Wikipedia has undeniable advantages as a source of information: it is free; it is available to anyone who has an Internet-connected computer or mobile phone; its information can be accessed easily and rapidly; it is familiar to almost every Internet user looking for information on general topics; and it includes links to other sources of information, among other factors [9,10]. A study examining the information available concerning diabetic retinopathy concluded that Wikipedia is the site with the best information, at least in English, receiving the highest score of all 11 websites examined, with a mean score representing 74% of total possible points [11].

English is the most widely used language in business, science, and many other fields of knowledge. However, only about 6% of the world's population are native English speakers, and 75% of people on the global level do not speak English at all [12]. The goal of this paper is to compare the completeness and quality of information about diabetic retinopathy on Wikipedia on the world's most commonly spoken languages in 2020, while also investigating whether the countries where the majority of patients reside have access to accurate and reliable information by using Wikipedia in their native language.

## Material and methods

An observational, descriptive, cross-sectional study. The quality and quantity of information on diabetic retinopathy obtained from the free encyclopedia Wikipedia® was assessed in languages with one hundred million or more total speakers (as a first or second language, L1 or L2) according to Ethnologue [13]. We also included six relevant languages due to their regional cultural weight and/or their tradition in medicine since ancient times: Greek, Italian, Korean, Turkish, Persian and Hebrew. This resulted in a total of 19 languages, which cover roughly three-quarters of humanity in terms of population:

- English

- German

- French

- Russian

- Spanish

- Italian

- Portuguese

- Chinese

- Japanese

- Korean

- Arabic

- Hindi

- Indonesian

- Urdu

- Bengali

- Turkish

- Persian

- Hebrew

- Greek

The information on the sites was collected by three ophthalmologist observers belonging to the Mexican Institute of Ophthalmology, who are native speakers of Spanish and fully fluent in English.

The "diabetic retinopathy" entry of the English version of Wikipedia was used as the starting point, and by using the "more languages" function, other languages were accessed. When unavailable, the term was translated using Google Translator into the desired language and then input into the search box of the corresponding version of Wikipedia. If not found, the entry was considered as void.

A database was created with the most important subtopics for the education of patients with diabetic retinopathy in any of its classifications, based on 25 questions. taken from the study by Kloosterboer et al. [11] and corroborated by experts from the Retina and Vitreous Service of the Mexican Institute of Ophthalmology, annexed in S1 Fig in S1 File.

The information was downloaded in February 2020, and a translation into English (including the Spanish version) was created in order to provide a clear basis for understanding by the three evaluators.

The results were stratified on a scale from 0 to 4. A result of 0 indicates no available information for that question; 1 point represents an answer that is unclear, inaccurate, which omits significant information or shows poor organization; 2 points reflect a partially complete item that somehow addresses the concept, but has gaps and unorganized information; 3 points are given when essential elements to answer the question are included and address the most relevant points in a focused and organized way; and 4 points correspond to a response that is accurate and complete, explaining the information in a clear, focused and organized manner [11].

Basic descriptive statistics with measures of the central tendency permitted assessment of the academic content of the information on diabetic retinopathy across languages, as well as the percentage obtained by each language compared with the theoretical maximum score on each topic.

The score obtained as a percentage of each language evaluated was compared. This is tabulated below in five ordinal categories (from 4 to 0) split in intervals of 25%: excellent (> 75%), fair (51–75%), substandard (26–50%), poor (1–25%), and no information at all (0%).

Comparative statistics were performed for correlation with the Spearman test and the Kappa coefficient, whose coefficient reflects the strength of agreement among all three observers, to assess the inter-observer variability between the three ophthalmologists.

## Results

The correlation test showed a Spearman concordance of 0.992 among the observers as well as a Kappa coefficient of 0.72, which is tabulated below; p-value was taken as significant, $<0.05$.

Nineteen Wikipedia entries corresponding to the 19 evaluated languages were reviewed, revealing the following scores (mean ± standard deviation (s.d.): English 73.00 ± 2.646, 12.5, German 66.67 ± 5.508, French 38.33 ± 4.163, Russian 30.67 ± 1.528, Spanish 26.00 ± 1.000, Italian 16.00 ± 5.292, Portuguese 23.67 ± 6.028, Mandarin Chinese 11.67 ± 1.155, Japanese 39.67 ± 5.508, Korean 3.00 ± 1.732, Arabic 17.00 ± 2.646, Indonesian 13.00 ± 2.646, Turkish 73.33 ± 3.215, and Persian 35.00 ± 1.732. Hindi, Urdu, Bengali, Hebrew, Greek had no information in Wikipedia on diabetic retinopathy as of December 2019. The overall results are summarized in Table 1.

The languages with percentages higher than 5% were registered in descending order based on their final mean score; and tabulated in five ordinal categories (excellent, fair, substandard, poor, no information); displayed in Fig 1.

No language achieved the label "excellent"; 3 languages were rated as "fair "; 4 languages qualified as "substandard"; and 7 languages were scored as "poor." No information could be found in five languages.

A verification was performed in January 2021 of the information, finding that 6 languages had undergone modification in their content, but this did not change the classification in which they were placed in 2019; the language that changed most was Chinese, where a difference of 9% was found, representing an improvement in quality.

The most often and least often answered questions in the 14 languages are also summarized in Table 2. The three most often answered were: 1) definition of the condition (all 14 languages) with an average score of 2.8 /4.0, and a percentage of 71% in the quality of the answer; 2) risk factors, (14 languages) with an average score of 2.50 / 4.00 (i.e. 63% of response quality)

**Table 1. Scores by language.**

|  | Minimum | Maximum | Median SD | Variance | % | Rating |
|---|---|---|---|---|---|---|
| English | 71 | 76 | 73.00 ± 2.65 | 12.5 | 70% | Fair |
| German | 64 | 73 | 66.67 ± 5.51 | 40.5 | 63% | Fair |
| French | 37 | 43 | 38.33 ± 4.16 | 18.0 | 37% | Substandard |
| Russian | 29 | 31 | 30.67 ± 1.53 | 2.0 | 29% | Substandard |
| Spanish | 25 | 27 | 26.00 ± 1.00 | 2.0 | 25% | Poor |
| Italian | 18 | 20 | 16.00 ± 5.29 | 2.0 | 15% | Poor |
| Portuguese | 18 | 23 | 23.67 ± 6.03 | 12.5 | 23% | Poor |
| Chinese | 11 | 13 | 11.67 ± 1.15 | 2.0 | 11% | Poor |
| Japanese | 34 | 40 | 39.67 ± 5.51 | 18.0 | 38% | Substandard |
| Korean | 2 | 5 | 3.00 ± 1.73 | 4.5 | 3% | Poor |
| Arabic | 14 | 18 | 17.00 ± 2.65 | 8.0 | 16% | Poor |
| Indonesian | 12 | 16 | 13.00 ± 2.65 | 8.0 | 12% | Poor |
| Turkish | 71 | 77 | 73.33 ± 3.21 | 18.0 | 70% | Fair |
| Persian | 34 | 34 | 35.00 ± 1.73 | 0.0 | 33% | Substandard |

Hindi, Urdu, Bengali, Hebrew, Greek: no information was obtained from Wikipedia.

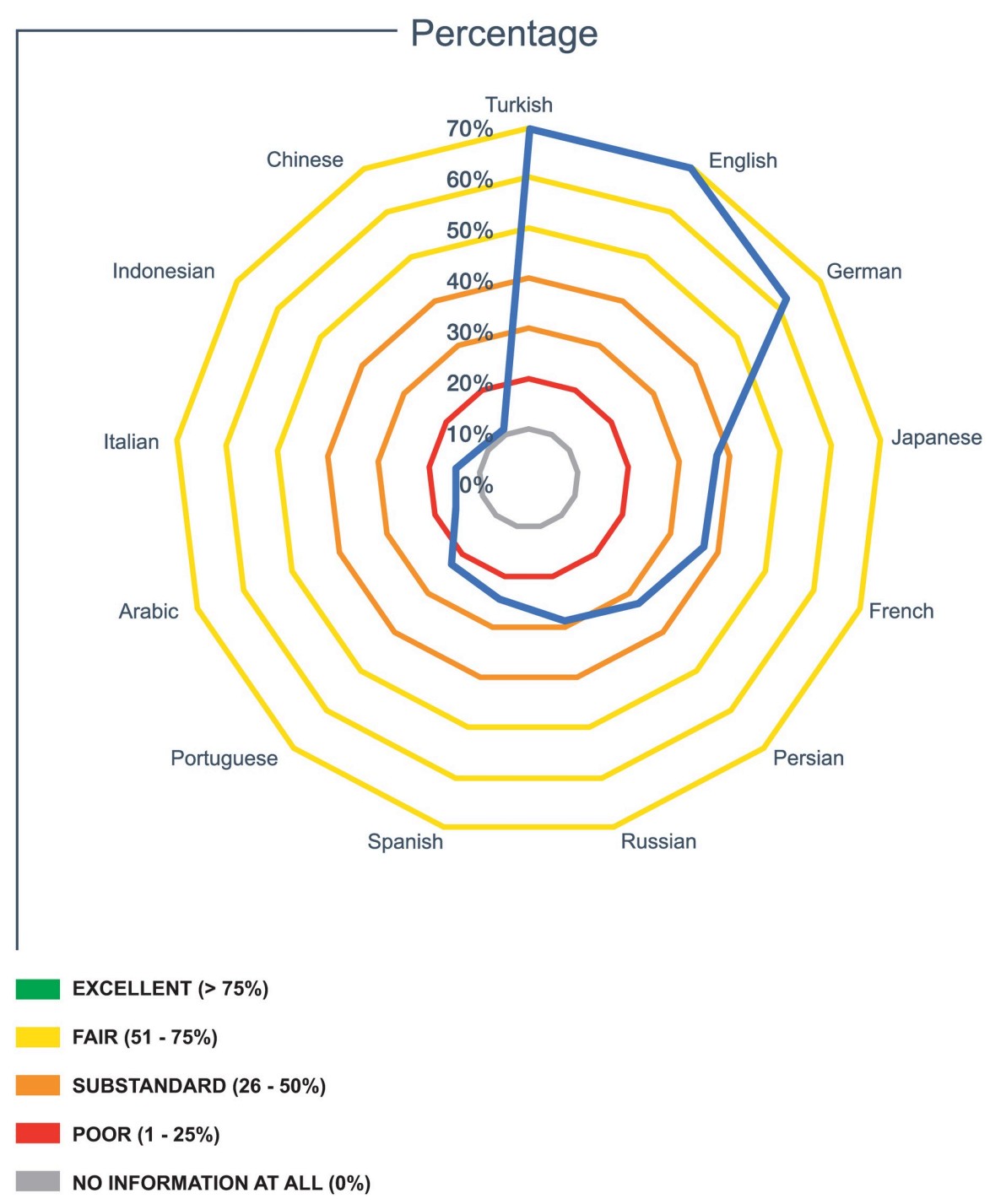

**Fig 1. Average percentage per language.**

and 3) symptoms (13 languages) with an average response of 2.36 / 4.00 (59% quality). The three least frequently answered were: 1) can visual acuity loss be reversed? (4/14), an average of 0.57 / 4.00 (14% of quality), 2) are there any telemedicine options available? (3/14) with an average of 0.43 / 4.00 (11% of quality) and finally 3) which is the most affected gender? (only 2/14) with an average score of 0.14 / 4.00 (4% of quality).

**Table 2. Question form for evaluation of information in each language; this represents the survey that was conducted on each Wikipedia page, as well with the number of pages that answered the question, its average score and corresponding percentage for each question in all the languages that were addressed.**

| Question | Pages that answered that question | Average score on all pages (0–4) | Percentage obtained in each question |
|---|---|---|---|
| What is diabetic retinopathy? | 14 | 2.86 | 71% |
| What are the symptoms of diabetic retinopathy? | 13 | 2.36 | 59% |
| What is the difference between nonproliferative and proliferative diabetic retinopathy? | 13 | 2.21 | 55% |
| How is diabetic retinopathy diagnosed? | 12 | 1.86 | 46% |
| Do I have to go to the eye clinic to be diagnosed or are there options for telemedicine screening? | 3 | 0.43 | 11% |
| When should screening start? | 7 | 0.86 | 21% |
| Once diagnosed, how often should I see my doctor for diabetic retinopathy? | 7 | 1.21 | 30% |
| What are the risk factors for diabetic retinopathy? | 14 | 2.50 | 63% |
| Can anything be done to reverse diabetic retinopathy? | 4 | 0.64 | 16% |
| What percentage of patients become legally blind from diabetic retinopathy? | 6 | 1.21 | 30% |
| How can vision loss be prevented? | 12 | 2.21 | 55% |
| Is vision loss reversible? | 4 | 0.64 | 16% |
| How is diabetic retinopathy treated? | 9 | 1.86 | 46% |
| What is panretinal photocoagulation, and what are the complications associated with it? | 10 | 1.57 | 39% |
| What is an anti-VEGF injection and what are the complications associated with anti-VEGF therapy? | 6 | 1.00 | 25% |
| Are anti-VEGF injections or laser a cure or do they need to be repeated? | 6 | 0.64 | 16% |
| What are the surgical treatments for diabetic retinopathy and what are the potential complications? | 6 | 1.00 | 25% |
| What is tractional retinal detachment? | 9 | 1.36 | 34% |
| What is diabetic macular edema? | 10 | 1.14 | 29% |
| What is a fluorescein angiogram? | 8 | 1.00 | 25% |
| What is optical coherence tomography? | 4 | 0.64 | 16% |
| Are there any oral medications that can alter the progression of diabetic retinopathy? | 10 | 1.29 | 32% |
| What is the incidence of diabetic retinopathy? | 11 | 1.93 | 48% |
| Which gender is most affected by diabetic retinopathy? | 2 | 0.14 | 4% |
| Which age group is most affected by diabetic retinopathy? | 5 | 1.07 | 27% |

## Discussion

In 2019, the countries with the highest number of adults with diabetes aged between 20 and 79 years were China (116.4 million), India (77 million) and the United States of America (31 million), and they are expected to remain the leaders through 2030.[21] The importance of having a free, accurate and reliable source of information on DR online for such a considerable number of patients is evident.

There are some 1 billion native speakers of Mandarin Chinese (including Standard Chinese) [13], of whom 116.4 million have diabetes (between 20–79 years), while more than 35 million might be affected by diabetic retinopathy. Many of these patients may be in need of information about their disease, care, treatment, and even prognosis. What is worrisome about this situation is that these patients will find incomplete and unreliable information of poor quality on a condition that is one of the leading causes of blindness worldwide. As shown in this study, the information about diabetic retinopathy on the Wikipedia platform in the

Chinese language is inadequate, with only 11.67 points out of a possible 105, according to our survey. This could leave patients with unanswered questions, unable to gain a clear understanding about the severity of the disease, therefore leading to poor disease control and a situation in which care is more focused on treating complications rather than being preventative.

The second leading language with the most native speakers is Spanish, with more than 460 million [13], and Mexico has the highest number of Spanish-speaking diabetics between the ages of 20 and 79. Among Spanish-speaking countries, only Spain and Equatorial Guinea are outside the Americas, and the most affected territories are Central and South America. For this region, a prevalence of 9.4% is reported for diabetes, similar to the global rate of 9.3% [5] observed across most countries with mid-to-upper levels of income [14]. Recommendations have been made for DR in terms of detection, referral, follow-up schedules, and types of treatment in environments with high, low, or intermediate levels of resources, classified in general terms according to the country's income level [15]. This directly affects the availability of information, preventive services, and treatment of the condition. Of the entire Spanish-speaking population, we estimate that 43 million people could be affected by diabetes, while up to 15 million have some degree of diabetic retinopathy. These patients will unfortunately find mediocre, unclear and incomplete information on their condition, with an average score of 26 points.

The situation is completely different for the world language with the third highest number of native speakers. There are 379 million people whose mother tongue is English [13], among whom the majority live in high-income countries such as Australia, Canada, the United Kingdom and the United States of America [16]. Additionally, if all speakers using it as a second language (L2) are counted, English becomes the leading language, with > 1.1 billion [13] This might encourage some Internet users to use it as a pivotal *lingua franca* if clear and precise information is unavailable in their own language.

In the USA alone there are 31 million people between 20 and 79 years of age with a diagnosis of diabetes, making it the country with the third highest number of diabetics worldwide [5] and in first place among English-speaking countries; no other English-speaking countries appear in the top ten with regard to the number of diabetics. Diabetic retinopathy is a leading cause of new cases of legal blindness among working-age Americans. The prevalence rate of retinopathy for all adults with diabetes age 40 and older in the United States is 28.5% (4.2 million people). It is likely that superior access to information technology, world-class medical centers, and the use of English as the "language of science" explains why English was the highest rated along with Turkish among the 19 languages included in this study.

The fourth and fifth leading languages in terms of native speakers worldwide are Hindi with 341 million and Bengali with 228 million speakers respectively. They are the mother tongues of India and Bangladesh, which are classified as low-middle income and have a large number of inhabitants affected by diabetes: India with 77 million (number two in the world) and Bangladesh with 8.4 million (tenth in the world) [5,13,17]. Using these estimates for both languages, approximately 8 million diabetics could be suffering from some degree of diabetic retinopathy, and they unfortunately do not have the opportunity to obtain answers to their questions through Wikipedia in their languages due to the lack of information on this platform, which is alarming given the large number of people affected.

The rating achieved in the Turkish language, which is in position number 13 with 79 million native speakers worldwide, is a surprising finding of our research. Turkey belongs to the category of upper-middle income countries and has a prevalence of 12% of diabetics between the ages of 20 and 79, giving a total of approximately 6.5 million diabetics, among whom approximately 2.2 million will have some degree of retinopathy. We are pleased to note that they can find information of higher quality and wider coverage than, for example, in Russian,

which has almost twice as many speakers, or Portuguese and Japanese, which in addition to having many more speakers are languages of high-income countries. The information in Turkish was rated at a level similar to that in English in terms of quality [5,13,16].

Official languages of high-income countries with a considerable number of speakers are not limited to Japanese, French or Russian: Korean also scored fair to poor in ratings of the information provided in that language. This is particularly surprising, since diabetes and DR are a public health problem in high-income countries as well, and education about these conditions should be a high intervention priority for all countries and regions [18].

Analyzing the two highest scores, with Turkish performing fairly well at the same level as the English language, we found that the contents in Turkish are a duplicate of the information presented on the English version of Wikipedia. We were not able to rate the quality of the translation into Turkish in terms of grammar and style, but this leads us to a common-sense remark. If no information or only poor information is present in a given language, why should Wikipedia editors in those languages not simply translate the best version from another Wikipedia? Considering that this information is in the public domain and there are gaps in knowledge, why should they wait until some user creates "original" content, if that subject is supported by evidence-based content in another version (language)?

On the other hand, we must emphasize that Wikipedia provided at best 74% of the information required by patients, in our professional estimate. The available content should be improved, not only on this platform, but also on other resources usually found by patients when searching for medical information.

The information found on the Internet needs to be both complete and understandable for users. It has been established that medical educational material for patients should have a readability level between the fourth and sixth grades of primary school [19–22], and the more effectively that patients are able to gain a good understanding of their medical condition, the better their treatment and follow-up will be, with reduced negative economic impact on their society [23].

As specified by Wikipedia, all entries have to be supported by sufficient references (i.e. specifying their source), allowing users to check for accuracy, precision or neutrality or even further completion of the information. As Wikipedia is not a primary source, information must be sourced from somewhere else, rendering referencing desirable.

We found that in the review one year later, the content of the pages did not undergo major changes, demonstrating that the Wikipedia reviewers have not been able to substantially improve the quality or quantity of the information for patients.

The availability and quality of information for patients can be a factor that facilitates an earlier diagnosis and better follow-up, and thus positively influences the prevention, treatment and prognosis of the disease. A final query might be addressed to ophthalmological and endocrinological societies of countries whose languages performed poorly in our study: if anyone can edit Wikipedia, should not these professionals do so?

The International Council of Ophthalmology gives us the number of ophthalmologists worldwide [24]. Considering that these specialists who speak Korean, French, Russian number more than 3,000, 8,000 and 14,000, respectively, one would surely conclude that half a dozen in each country could be found to write a Wikipedia article on a key subject during a weekend session. There are 15,000 ophthalmologists in India and only 600 in Bangladesh, but we also presume that a translation project could be undertaken by a few of them and would be considered as a service to their communities.

It has recently been shown that Wikipedia is a prominent health information resource for the public or patients seeking health information online. Wikipedia's health content is accessed frequently, and its pages regularly rank highly in Google search results [25]. Problems

of Internet access in remote areas or poorer countries are beyond the scope of this paper and affect all other websites, but access to the Web is dramatically increasing everywhere.

We must acknowledge the weaknesses of Wikipedia, especially that it can be edited by anyone with a registered account, but this does not detract from the fact that internet users consult Wikipedia as one of their main methods of data retrieval [11]. In spite of its shortcomings, its wide use proves it cannot be simply ignored by health professionals, and precisely because we cannot prevent the public from accessing this information, our call is for health institutions from countries where information is lacking or incomplete to consider their social responsibility and use this already existing free channel to provide high-quality information to patients and other persons interested in the topic.

## Conclusions

As would be expected, the quality of content related to diabetic retinopathy on Wikipedia is variable across different languages. However, it is unfortunate that many culturally and numerically important languages cannot provide adequate information to the lay public on this topic on Wikipedia. However, we note that if anyone can edit Wikipedia, general practitioners and ophthalmologists can do so as well. Our findings suggest that ophthalmologists in countries where the score in our study is low might fruitfully devote some time to provide better content.

In places with a lively academic life, like Korea, original texts created by local physicians are not an unreasonable expectation. In countries where doctors exist in fewer numbers, the translation of a version that is quite good, accurate, and complete is not a time-consuming task. Specialists from many former British territories which were found to have lower-quality information in this study, such as India and Bangladesh, have doctors with excellent or fluent English, and this translation task should not be difficult due to their obvious mastery of their native tongues, not even considering the diaspora in Western countries which constitutes an elite who could partially pay back in this manner the benefit of undergraduate medical education they received in their homeland. They could even start with the most basic information, further expanding it as time allows. As an aside, it is likely that some of these languages lack accurate words or phrases for some specialized medical terms, and thus a Wikipedia initiative could provide a lasting contribution by creating or standardizing such vocabulary.

The Turkish version of Wikipedia, which is an adaptation of the English version, is a clear demonstration that simple initiatives can greatly improve the knowledge made available to the layperson on the Internet on any medical topic, diabetic retinopathy in this case. We hope that scientists from other areas will explore this topic in their own area of expertise.

## Supporting information

**S1 File.**
(DOCX)

## Author Contributions

**Conceptualization:** Kouatzin Aguilar-Morales, Gustavo Aguirre-Suarez, Angel Lee, Van Charles Lansingh.

**Data curation:** Kouatzin Aguilar-Morales, Gustavo Aguirre-Suarez.

**Formal analysis:** Kouatzin Aguilar-Morales, Gustavo Aguirre-Suarez, Angel Lee.

**Funding acquisition:** Van Charles Lansingh.

**Investigation:** Kouatzin Aguilar-Morales, Gustavo Aguirre-Suarez, Van Charles Lansingh.

**Methodology:** Kouatzin Aguilar-Morales, Gustavo Aguirre-Suarez, Angel Lee.

**Project administration:** Kouatzin Aguilar-Morales, Angel Lee, Van Charles Lansingh.

**Resources:** Van Charles Lansingh.

**Supervision:** Kouatzin Aguilar-Morales, Gustavo Aguirre-Suarez, Angel Lee, Van Charles Lansingh.

**Validation:** Kouatzin Aguilar-Morales, Gustavo Aguirre-Suarez, Brigham Bowles, Angel Lee, Van Charles Lansingh.

**Visualization:** Kouatzin Aguilar-Morales, Gustavo Aguirre-Suarez, Brigham Bowles, Angel Lee.

**Writing – original draft:** Kouatzin Aguilar-Morales, Gustavo Aguirre-Suarez, Angel Lee.

**Writing – review & editing:** Kouatzin Aguilar-Morales, Brigham Bowles, Angel Lee, Van Charles Lansingh.

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
