## [Decision Letter · Decision Letter 0]

22 Jul 2021

PONE-D-21-08214

Wikipedia, friend or foe regarding information on diabetic retinopathy? A content analysis in the world’s leading 19 languages

PLOS ONE

Dear Dr. Van Lansingh,

Thank you for submitting your manuscript to PLOS ONE. After careful consideration, we feel that it has merit but does not fully meet PLOS ONE’s publication criteria as it currently stands. Therefore, we invite you to submit a revised version of the manuscript that addresses the points raised during the review process.

We look forward to receiving your revised manuscript.

Kind regards,

Rohit C. Khanna, MD, MPH

Academic Editor

PLOS ONE

Journal Requirements:

**3. **Please include a copy of Table 3 which you refer to in your text on page 10.

Additional Editor Comments (if provided):

One of the major issue is that the paper does not mention how information is collected and written in Wikipedia i.e. the functional model of Wikipedia. Unless the methodology as well as limitations of Wikipedia are known, it’s not fair to give recommendations

Apart from this, most of the elderly population still do not access Wikipedia and some have blocked it (China, Myanmar etc). What percentage of population above 40 years have access to Wikipedia? So will the information reach other group of individuals who don’t have access?

Computerized translations are not perfect. So how one can assess if the same information is communicated in thee translated language?

Reviewers' comments:

Reviewer's Responses to Questions

**Comments to the Author**

1. Is the manuscript technically sound, and do the data support the conclusions?

Reviewer #1: Yes

Reviewer #2: No

Reviewer #3: Yes

Reviewer #4: Yes

Reviewer #5: Partly

Reviewer #6: Yes

2. Has the statistical analysis been performed appropriately and rigorously? 

Reviewer #1: Yes

Reviewer #2: Yes

Reviewer #3: Yes

Reviewer #4: N/A

Reviewer #5: N/A

Reviewer #6: No

3. Have the authors made all data underlying the findings in their manuscript fully available?

Reviewer #1: Yes

Reviewer #2: Yes

Reviewer #3: Yes

Reviewer #4: Yes

Reviewer #5: Yes

Reviewer #6: Yes

4. Is the manuscript presented in an intelligible fashion and written in standard English?

Reviewer #1: Yes

Reviewer #2: Yes

Reviewer #3: Yes

Reviewer #4: Yes

Reviewer #5: Yes

Reviewer #6: Yes

5. Review Comments to the Author

Reviewer #1: Thank you for submitting your paper to this journal. The only concern I have is the use of "most important languages". For everyone, their mother tongue will be the most important and so this might be misunderstood by some. If you say most prominent/spoken by most people, then you do not pass judgement on where they stand in terms of importance.

Reviewer #2: Aguilar-Morales et al assessed whether or not Wikipedia was a valid source of information for patients interested in learning about diabetic retinopathy.

I have several observations regarding their manuscript

1- They assume that internet access is the same worldwide.

" China. ... As of April 2019, all versions of Wikipedia are blocked in China. The Chinese Wikipedia was launched in May 2001. Wikipedia received positive coverage in China's state press in early 2004 but was blocked on 3 June 2004, ahead of the 15th anniversary of the 1989 Tiananmen Square protests."

https://en.wikipedia.org/wiki/Censorship_of_Wikipedia. (Accessed on June 18 2021)

2- Computerized translations are not perfect ... how did they actually translate the contents of each language? What languages are the authors fluent in ?

3- They assume that patients invariably will choose Wikipedia as their number one source of information. So for instance if I live in India and I search for "diabetic retinopathy" what will such a search reveal? Will wikipedia be in the top hits?? I just searched for the term "retinopatia diabetica" on google in Costa Rica and the wikipedia site is NOT even in the first ten pages!! Top hits include the mayo clinic, american academy of ophthalmology and the national eye institute.

In summary, perhaps their assessment that wikipedia is not an excellent source of information may be true but hardly relevant.

Reviewer #3: It is an interesting paper!

• In countries such as India, English is often used in the internet search though it is not the first language. Hindi is not often used for internet search though it happens to be the national language. Same may be true for other languages such as Bengali.

• Proportion of people speaking English in addition to other major languages listed may provide more information. Authors can consider adding this information, it this possible.

• Authors can mention the limitation of the approach used in the paper.

Reviewer #4: This is a very original paper in a very important disease. The topic of the paper give us a new view of of information is delivered to the public and how it is managed. The paper question how trustable is the information from wiki sources.

Reviewer #5: This is an interesting article.

However, patients get information from Wikipedia as an important source is questionable. Especially when we know that the content does not need to be posted by physicians/healthcare professionals.

How did the authors analyze and secure the information related to different languages? Did they use a translator? All of these points can also create a bias.

Unless this article information can be forwarded to Wikipedia and some changes can be made to change the content, the article exercise is not useful.

Is there any way to force that change in Wikipedia?

Reviewer #6: Overview:

The subject of the manuscript is very interesting and it is worth being analyzed and disseminated. It is important to evaluate the level of accuracy of the Diabetic Retinopathy information presented in the principals’ world websites. These websites could help diabetes patients to make medical decisions for their treatment or prevention. Although this article is clear and professionally written, my mains concerns are in the statistics used and some recommendation in the discussion section.

Specifics concerns:

· The table 1 should be in the result section.

· Regarding the “Average score on all pages (0-4)” presented in the table 1, I recommend treating it as a categorical or ordinal variable. Presenting in table 1 the frequency and the percentage of each of these items (0, 1, 2, 3 and 4).

· In line 130: the results of the Spearman and Kappa analysis should be in the result section.

· I recommend checking the analysis in table 2, because it's not common than the range of the 95% IC (Inf. – Sup.) is bigger than the range (Min. – Max.) of all the data.

· In the part where is discussed about the Mandarin Chinese population, would be interesting to explain the impact of the blocking of Wikipedia’s website in this country.

· India, a very populated country had internet access to the 50% of their population, could be interesting to assume the Wikipedia education impact under this situation.

· Finally, I recommend a bibliography revision about how online education improve the prevention, treatment and prognostic in diabetes or sick patients. Could be interesting to evaluate the advantages, opportunities, threats and barriers that exist in online education.

6. PLOS authors have the option to publish the peer review history of their article (what does this mean?). If published, this will include your full peer review and any attached files.

Reviewer #1: **Yes: **Tunde Peto

Reviewer #2: No

Reviewer #3: No

Reviewer #4: No

Reviewer #5: No

Reviewer #6: No

---

## [Author Response · Author response to Decision Letter 0]

1 Sep 2021

Dear Editor and Reviewers, 

Thank you very much for your comments and corrections. We are adding a letter here to explain each point.

We have already checked again that it fulfills the requirements of the journal, as well as the order and format.

Reviewer #1: Thank you for your comments. We have modified the objective of the study with the following: To compare the completeness and quality of information about diabetic retinopathy on Wikipedia in the most spoken languages in the world.

Reviewer #2: Thank you for taking the time to reply to us. With specific regard to the questions we would like to comment that, despite the restriction of Wikipedia in China, we did find a Wikipedia page in that language in the mentioned topic, because we do not exclude Chinese speakers outside the online restrictions of the country. 

We explain in the article that our mother tongue is Spanish, but three co-authors are fluent in English, so we were able to understand the literal translations of the content in each language. 

In the searching of the term, the results depend on the algorithm of the preferred browser, in this case Google of the previous research to give more consistent results, for example, if we physicians enter a term is more likely to yield scientific pages, but for the average user often the first option in addition to the fact that it is the majority of people are familiar with (not only for medical terms) and is the most widespread, In the articles Kloosterboer A et al 2019 compared the quantity and quality of information on this issue on various websites including (American academy of ophthalmology, Mayo clinic, and others) and concluded that for patient understanding, Wikipedia was the most complete and clearest as well as accurate for the term.

Reviewers #3 and #5: Thank you for your feedback. We already specified in the text and took into account the total number of English speakers including those of us who have English as a second language. We have also included a paragraph with the vulnerabilities and limitations of the study.

Reviewer #4 and #5: Thanks for the great comments; according to Wikipedia´s rules the articles have to have mandatory references for publication as well as a review by the editors of this online encyclopedia. As for the secondary objective, it is to propose that experts on a certain subject should be responsible for modifying the information that is posted on Wikipedia so that it can be updated from a certain time; this is in the discussion section.

Reviewer #6: Thank you very much for the revisions; we recognized the error in the confidence intervals and we corrected them, as well as the order of the tables and paragraphs.

---

## [Decision Letter · Decision Letter 1]

23 Sep 2021

Wikipedia, friend or foe regarding information on diabetic retinopathy? A content analysis in the world’s leading 19 languages

PONE-D-21-08214R1

Dear Dr. Van C Lansingh

We’re pleased to inform you that your manuscript has been judged scientifically suitable for publication and will be formally accepted for publication once it meets all outstanding technical requirements.

Kind regards,

Rohit C. Khanna, MD, MPH

Academic Editor

PLOS ONE

Additional Editor Comments (optional):

Reviewers' comments:

Reviewer's Responses to Questions

**Comments to the Author**

1. If the authors have adequately addressed your comments raised in a previous round of review and you feel that this manuscript is now acceptable for publication, you may indicate that here to bypass the “Comments to the Author” section, enter your conflict of interest statement in the “Confidential to Editor” section, and submit your "Accept" recommendation.

Reviewer #1: All comments have been addressed

Reviewer #5: All comments have been addressed

Reviewer #6: All comments have been addressed

Reviewer #7: All comments have been addressed

2. Is the manuscript technically sound, and do the data support the conclusions?

Reviewer #1: Yes

Reviewer #5: Yes

Reviewer #6: Yes

Reviewer #7: Yes

3. Has the statistical analysis been performed appropriately and rigorously? 

Reviewer #1: Yes

Reviewer #5: N/A

Reviewer #6: Yes

Reviewer #7: N/A

4. Have the authors made all data underlying the findings in their manuscript fully available?

Reviewer #1: Yes

Reviewer #5: Yes

Reviewer #6: Yes

Reviewer #7: Yes

5. Is the manuscript presented in an intelligible fashion and written in standard English?

Reviewer #1: Yes

Reviewer #5: Yes

Reviewer #6: Yes

Reviewer #7: Yes

6. Review Comments to the Author

Reviewer #1: Thank you for answering my questions. The paper benefited from the changes and from my point of view, it will contribute to science.

Reviewer #5: Present conclusion is not clear. Conclusion to be rewritten answering the research question (title) of the manuscript ( Wikipedia friend/ foe ?)

Reviewer #6: The subject of the manuscript is very interesting and it is worth being analyzed and disseminated. It is important to evaluate the level of accuracy of the Diabetic Retinopathy information presented in the principals’ world websites. This article is clear and professionally written.

Reviewer #7: The authors have addressed the comments provided in the initial review. It would have been useful if authors had provided point to point response to the reviewers comments.

7. PLOS authors have the option to publish the peer review history of their article (what does this mean?). If published, this will include your full peer review and any attached files.

Reviewer #1: No

Reviewer #5: No

Reviewer #6: **Yes: **Jaime Soria Viteri

Reviewer #7: No

---

## [Editor Report · Acceptance letter]

20 Oct 2021

PONE-D-21-08214R1 

Wikipedia, friend or foe regarding information on diabetic retinopathy? A content analysis in the world’s leading 19 languages 

Dear Dr. Lee:

I'm pleased to inform you that your manuscript has been deemed suitable for publication in PLOS ONE. Congratulations! Your manuscript is now with our production department. 

Kind regards, 

on behalf of

Dr. Rohit C. Khanna 

Academic Editor

PLOS ONE